# Model-Based Decision Support System for Electric Arc Furnace (EAF) Online Monitoring and Control

Bernd Kleimt [1,*], Waldemar Krieger [1], Diana Mier Vasallo [2], Asier Arteaga Ayarza [2] and Inigo Unamuno Iriondo [3]

[1]   VDEh-Betriebsforschungsinstitut GmbH, Sohnstrasse 69, 40237 Duesseldorf, Germany; waldemar.krieger@bfi.de
[2]   Sidenor I+D S.A., 48970 Basauri, Spain
[3]   Sidenor Aceros Especiales, 48970 Basauri, Spain
*   Correspondence: bernd.kleimt@bfi.de

**Abstract:** In this work, a practical approach for a decision support system for the electric arc furnace (EAF) is presented, with real-time heat state monitoring and control set-point optimization, which has been developed within the EU-funded project REVaMP and applied at the EAF of Sidenor in Basauri, Spain. The system consists of a dynamic process model based on energy and mass balances, including thermodynamic calculations for the most important metallurgical reactions, with particular focus on the modelling of the dephosphorisation reaction, as this is a critical parameter for production of high-quality steel grades along the EAF process route. A statistical scrap characterization tool is used to estimate the scrap properties, which are critical for reliable process performance and accurate online process control. The underlying process models and control functions were validated on the basis of historical production and measurement data of a large number of heats produced at the Sidenor plant. The online implementation of the model facilitates the accurate monitoring of the process behaviour and can be applied for exact process end-point control regarding melt temperature as well as oxygen, carbon and phosphorus content. Embedded within a model predictive control concept, the model can provide useful advice to the operator to adjust the relevant set-points for energy and resource-efficient process control.

**Keywords:** electric arc furnace; process models; process control; scrap characterisation; scrap mix optimisation





## 1. Introduction

The electric arc furnace (EAF) is the most important aggregate for steelmaking by recycling of secondary raw materials. The scrap used as a charge material in the EAF is characterised by a high variability in metallic yield, chemical composition and melting behaviour. This is essential to be considered in context with online monitoring and control of the EAF process. Within the framework of the Horizon 2020 project "Retrofitting Equipment for Efficient Use of Variable Feedstock in Metal Making Processes" (REVaMP), for the steel use case at Sidenor Aceros Especiales in Spain, a decision support tool was developed, which allows the operator, on the one hand, to characterize the properties of the different scrap types in use and, on the other hand, to use this information for scrap mix optimization as well for model-based monitoring and control of the EAF process. The overall target of this decision support tool is to reduce the overall energy consumption while reliably achieving the target values of the EAF process, and to facilitate as much as possible the use of low-quality scrap types for the production of high-quality steel grades. Both aspects contribute to the aims of the Green Deal of the EU, reducing the $CO_2$ emissions of steel production and at the same time strengthening the circular economy. The development of this decision support tool was based on previous work of the authors in the two areas described below.

A statistical modelling approach for characterising the properties of the steel scrap in use by means of a multi-linear regression was first developed within the project FLEX-CHARGE [1], which was funded in the European RFCS steel research fund. Within this project, the first version of a scrap mix optimization software based on a simplex optimization was also developed. It is evident that a scrap mix optimization facilitates a reduction in EAF production costs and an improvement in steel quality and environmental impact at the same time [2,3]. The developed solutions were based on MATLAB stand-alone applications, whereas within the REVaMP project, a user-friendly web-based solution with direct database access was developed. Recently, this statistical approach was extended towards an online supervision of scrap properties to detect significant deviations in expected scrap quality on a short-term basis [4,5].

The dynamic EAF process model of VDEh-Betriebsforschungsinstitut BFI, which is based on analytical mass and energy balance calculations, was developed in a first version as part of the RFCS EAFDynCon project [6] for a DC EAF for use in carbon steel production. Besides its application at several AC EAFs with scrap-based carbon steel production, it has also been applied in an EAF with continuous feeding of DRI [7] and in a small stainless steelmaking plant [8].

Nowadays, such models are widely used for simulation, online monitoring and dynamic control of the EAF steelmaking process. A good overview on offline simulation models for the EAF is given in [9], a detailed simulation model is, e.g., described in [10]. The online application of dynamic EAF models is based on cyclic and acyclic measurements and on a real-time version of statistical or analytical process models, which are normally part of a level-2 automation system. Numerous dynamic models for online application have been developed for the EAF process based on analytical mass and energy balance models [11–14] and statistical models based, e.g., on neural networks and fuzzy logic [15–19]. The main focus of these models is to predict online the current status of the melt with respect to steel temperature as well as steel and slag composition, and to use this information, e.g., for end-point control. More recent applications are dealing with the issues of auto-calibration of model parameters [20,21] and the optimization of operating patterns for electrical and chemical energy input [22].

However, so far, these modelling approaches have not sufficiently addressed the prediction of the dephosphorisation reaction in the EAF, although this is becoming more important when also producing high-quality steel grades via the EAF steelmaking route. So far, mainly offline simulation models for EAF dephosphorisation [23,24] have been developed and validated with process data from single heats, and some strategies for an improved slag practice were derived. Several investigations performed outside Europe focused on dephosphorisation when using DRI as a charge material [25].

Within this paper, the results of the REVaMP project work for the steel use case are presented with respect to a web-based toll for scrap characterisation and a dynamic EAF process model for online monitoring and control, addressing in detail the dephosphorisation reaction in the EAF.

## 2. Materials and Methods

### 2.1. Characterisation of Steel Scrap

The fundamental raw material at the electric steelmaking plant of Sidenor is steel scrap from three different origins:

- Post-consumer scrap: Old scrap from the demolition of the metal structure of industrial buildings, machinery, railway and naval scrap, used cars, etc.;
- Pre-consumer scrap: Industrial or new scrap that is generated in processing industries that use steel as raw material in their manufacturing processes;
- Internal recoveries: Scrap generated along the steelmaking process itself, in melt shops, rolling mills and other processes inside Sidenor premises.

Pre-consumer scrap is usually very clean in its chemical composition, and its variability is lower compared to post-consumer scrap. However, its use is, due to its high price, mainly

limited to the production of high quality steel grades. Well managed, internal recoveries can also provide a stable chemical composition that can lead to huge savings on ferroalloys additions. The critical scrap types are the post-consumer scraps with only roughly known composition and high variability in their properties.

As a first step, multi-linear regressions were performed based on 2019 process data regarding used scrap types, achieved meltdown analysis and tap weight, with the objective of visualizing statistically the variability of the 11 scrap grades currently in use at Sidenor. In Figure 1 is shown as example of the copper content of each of the 11 scrap types, visualized as a so-called violin diagram with average contents and their statistical distributions. Similar figures were created for all other important elements included in the scrap, such as Mn, Cr, P, etc., as well as for the overall metallic yield.

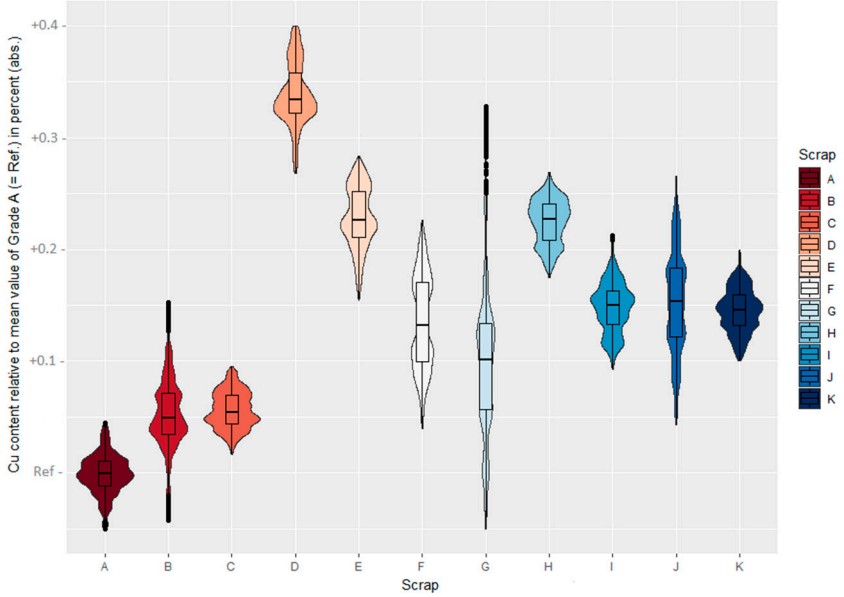

**Figure 1.** Copper content variability by scrap type at Sidenor.

In particular, the post-consumer scrap types D and E show high Cu contents with a large variability.

These investigations of Sidenor were the basis for VDEh-Betriebsforschungsinstitut (BFI) to develop a web-based tool for easy and comfortable assessment of scrap properties (metallic yield, element composition, specific meltdown energy requirement) on the basis of database access for the above-mentioned process data. Figure 2 shows exemplarily the Graphical User Interface (GUI) for selection of the evaluation period via heat numbers. A multi-linear regression calculation determines the properties of the different scrap types with mean value and standard deviation. The accuracy of the characterisation is visualised by a graph plotting the predicted versus the measured and analysed values of tap weight, steel analysis and electrical energy consumption, respectively. The latter one is shown in the graph of Figure 2b. Also, the error standard deviation of this prediction is provided.

In this web application, a scrap mix optimization tool is also included, which directly applies the currently identified characteristics of the scrap types in use to calculate the cost and quality optimal scrap mix for each steel grade to be produced [26]. The optimization allows us not only to minimize the scrap purchase costs, but also the costs for meltdown of the different scrap types, as a scrap type which may be cheap on the scrap market can turn out to be an expensive scrap type when considering the costs for the required meltdown energy.

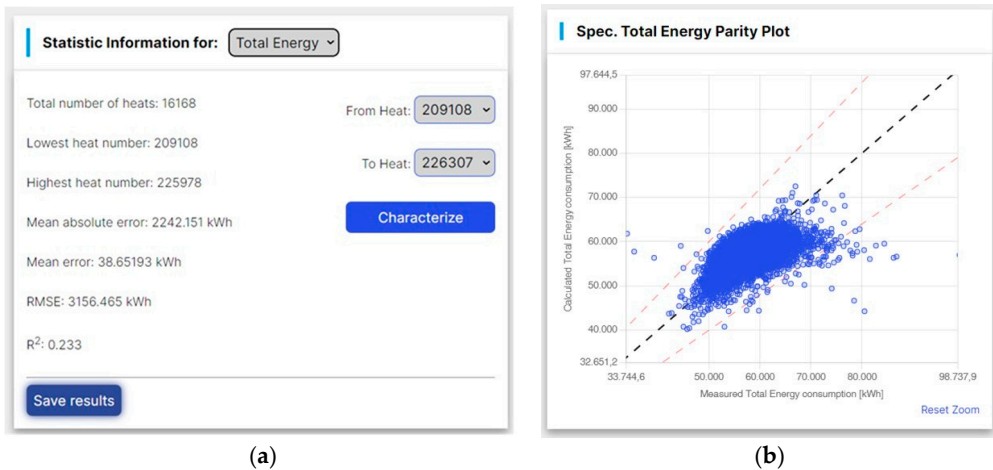

(**a**)                               (**b**)

**Figure 2.** GUI for selection of heats to be evaluated and accuracy of prediction (**a**) and evaluation results for electrical energy consumption (**b**) of the web-based tool for scrap type characterisation.

### 2.2. Dynamic EAF Process Model

The above-mentioned parameters for characterisation of the scrap types in use are also important inputs for a dynamic model for the EAF process in order to monitor and predict the process behaviour regarding the heat state with respect to the melt temperature as well as the composition of steel and slag.

This dynamic process model, which was developed by BFI in previous projects [5,27], describes the energetic status of the melt by dynamic mass and energy balance calculations, with integration of thermodynamic calculations for the most important metallurgical reactions like decarburisation and dephosphorisation. In Figure 3, the structure of the process model with the main function blocks, and the most important input values is shown.

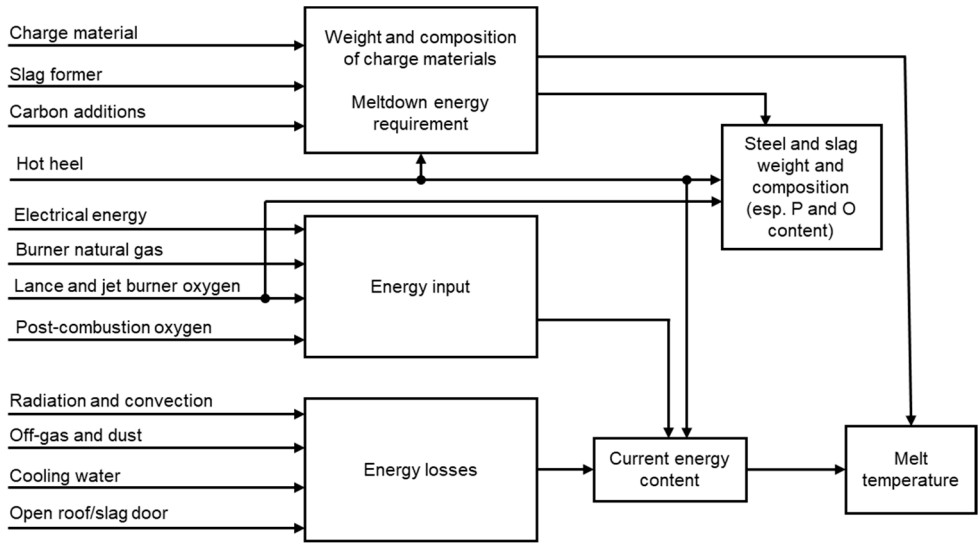

**Figure 3.** Structure of the BFI dynamic process model for the EAF.

The model calculates the total weight and composition (steel and slag) of the melt from the scrap, carbon and slag formers charged with the scrap basket, as well as carbon and lime injection via shell injectors. From the specific meltdown energy of each material, the total required energy is calculated. For the scrap types, the composition and meltdown energy requirement are determined using the above-mentioned tool for scrap characterisation. Within this, it is also considered that the non-metallic part of the scrap is added to the slag phase. The energy input comprises the electrical and the chemical energy input. The latter one consists of the energy input by burner gas and the oxygen input through door lances

and shell injectors and for post-combustion purposes. The Sidenor EAF is equipped with 3 burners, which can be operated in a jet mode to introduce oxygen into the steel bath, and 2 carbon lances for carbon supply and generation of a foamy slag. The energy losses comprise losses via off-gas, water cooling of panels and roof and overall radiation losses. The difference between energy inputs and losses gives the actual energy content of the melt, which relates to the meltdown energy requirement of the materials to calculate the current melt temperature.

The steel and slag composition are dynamically calculated by considering the different material inputs as well as chemical reactions due to oxidation and reduction via oxygen and carbon inputs, respectively. The rate of these oxidation and reduction processes is implemented proportional to the oxygen and carbon inputs. This approach considers that decarburisation can be performed directly through oxygen input via lances and injectors, but also indirectly via reduction of FeO in the slag. Thermodynamic equilibria are considered for elements where this simplified approach is not sufficient, e.g., for dephosphorisation.

For Sidenor, the phosphorus content plays a critical role to achieve the quality of the produced steel. Therefore, a special focus has been set on precise modelling of the phosphorus content during the EAF treatment. A mass transfer-limited approach was chosen with an equilibrium correlation proposed by Assis et al. [28] to describe the phosphorus distribution between steel and slag. Steel- and slag-side mass transfer are modelled via 1st order differential equations as follows

$$\frac{d[\%P]}{dt} = -\frac{A\rho_{St}k_{P,St}}{m_{St}}\left([\%P]-[\%P]^*\right) \tag{1}$$

$$\frac{d(\%P)}{dt} = -\frac{A\rho_{Sl}k_{P,Sl}}{m_{Sl}}\left((\%P)-(\%P)^*\right) \tag{2}$$

and coupled via the mass balance and partition ratio:

$$\frac{d[\%P]}{dt}m_{St} + \frac{d(\%P)}{dt}m_{Sl} = 0 \tag{3}$$

$$L_P = \frac{(\%P)^*}{[\%P]^*} \tag{4}$$

with $[\%P]$ and $(\%P)$ as phosphorus concentration in the steel and slag phase, $A$ as interfacial area for the reaction, $\rho$ as density, $k_P$ as mass transfer coefficient and m as mass. The indices *St* and *Sl* indicate steel and slag phase, respectively, and * is the equilibrium concentration at the interface. The phosphorus equilibrium concentration is derived from the phosphorus partition coefficient $L_P$, which is described by the following correlation [28]:

$$Log\left(\frac{f\cdot L_P}{Fe_t^{2.5}}\right) = 0.068[(\%CaO) + 0.42(\%MgO) + 1.16(\%P_2O_5) + 0.2(\%MnO)] + \frac{11570}{T} - 10.52 \tag{5}$$

It considers that the equilibrium conditions for dephosphorisation are favoured by a lower melt temperature, as well as by higher contents of *FeO*, *CaO* and *MgO* in the slag phase. Also, the effect of the concentrations of $P_2O_5$ and *MnO* in the slag phase is taken into account.

## 3. Results and Discussion

### 3.1. Validation of the EAF Process Model

The model parameters of the dynamic process model of BFI were tuned using acyclic and cyclic process data collected during the normal production of around 250 heats produced at the Sidenor EAF as input data. As such, a design of experiments was not applied for model validation. For validation, measurements regarding melt temperature and steel and slag composition were used. Model parameters are the corresponding scrap properties as well as energy efficiency, reaction, mass transfer and equilibrium parameters. In the

following, the validation results for the melt temperature and critical elements (carbon, oxygen, phosphorus) are presented with statistical information about all evaluated heats and the temporal development of the target variables in an example heat.

The calculated melt temperature results from the dynamic energy balance for the Sidenor EAF. To verify the dynamic process model, the calculated temperature is compared to the temperature measurements. In addition, the melting degree of the charged materials is calculated. In Figure 4, the accuracy for all evaluated heats is given, including mean value and standard deviation of the temperature prediction error.

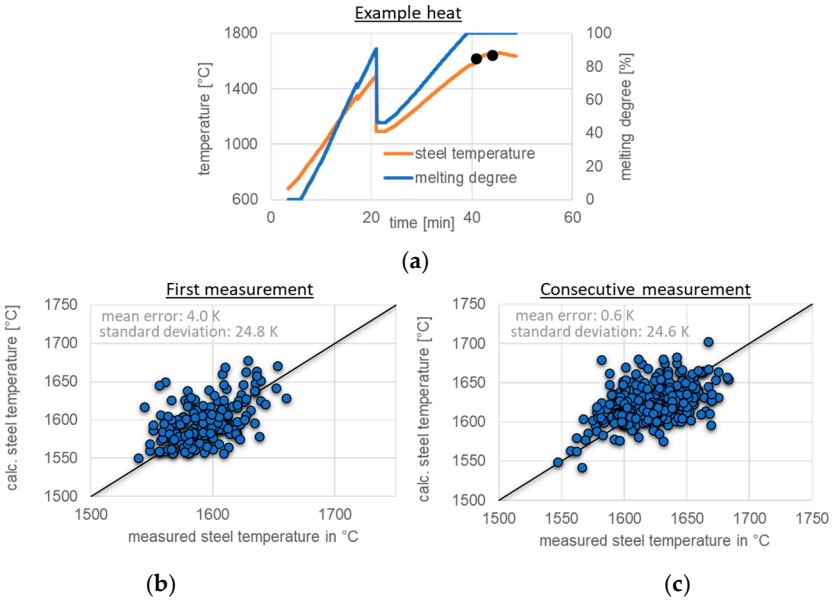

(a)

(b)                                                    (c)

**Figure 4.** Calculated steel melt temperature and comparison with measurements for one example heat: black dots indicate measurements (**a**); model accuracy of steel melt temperature calculation (**b**): first measurement; (**c**): further measurements after adaption to first one.

An error standard deviation of around 25 K was achieved, which is a good accuracy considering the energy throughput which has to be covered by the balance calculation. The error standard deviation of 25 K corresponds to an error of roughly 1% with respect to the energy balance calculation with an energy throughput of nearly 800 kWh/t of liquid steel. The inhomogeneity of the melt is a further significant source of scatter. This is proven by the fact that, sometimes, two consecutive temperature measurements with a time difference of only one minute can show temperature differences of 20 K.

The carbon concentration in the steel phase is calculated based on the material inputs and oxidation caused by the jet oxygen. Oxygen concentration in the melting phase is based on the equilibrium with carbon, and during the refining phase, the driving force for oxygen accumulation is defined by the difference between maximum solubility of oxygen and current oxygen concentration. Further, oxygen is used for direct reaction with the carbon injected via the carbon lances. In Figure 5, the development of carbon and oxygen content for an example heat (a) and the accuracy for all evaluated heats (bottom) is depicted. Note that the oxygen concentration is measured together with temperature, and therefore, more data points for oxygen are presented. The dynamic process model adapts to the measured oxygen value to increase the accuracy for further calculations, similarly to the temperature modelling. The carbon and oxygen predictions are both very accurate for the investigated heats, when considering that, for example, the carbon content of the charged scrap types can only be roughly determined. The oxygen concentration in the steel is an especially important quality indicator for Sidenor, and thus, it is important to provide an accurate online calculation to reduce the number of required measurements.

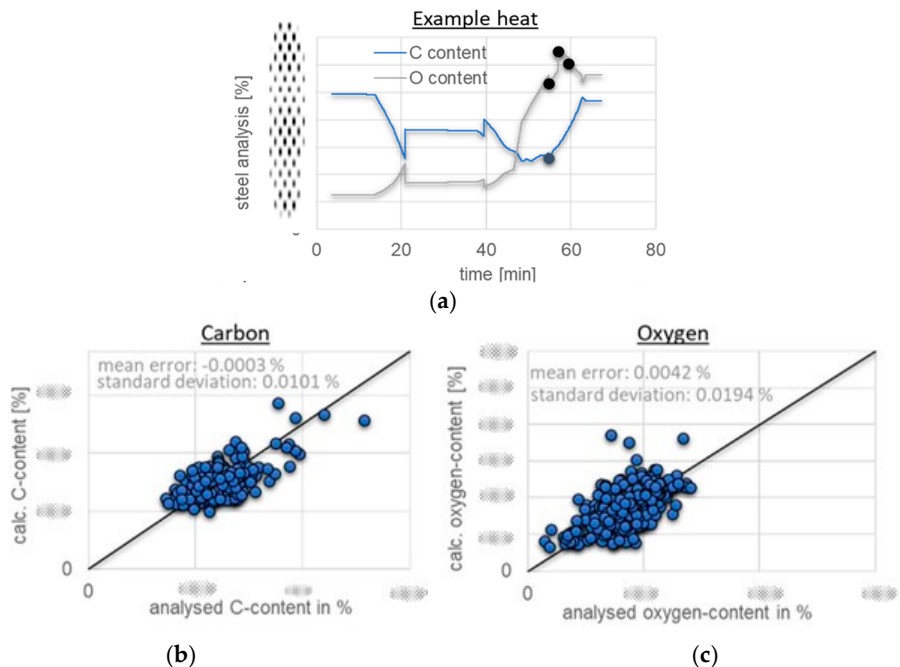

**Figure 5.** Calculated carbon and oxygen content and comparison with measurements for one example heat (**a**); model accuracy of carbon and oxygen calculation ((**b**): carbon; (**c**): oxygen).

As described above, the phosphorus concentration is calculated as mass transfer limited process based on equilibrium data between steel and slag phase. The equilibrium behaviour is mostly seen towards the end of the EAF treatment as the temperature decreases slightly, which shifts the equilibrium towards the steel side and results in the increase in phosphorus concentration (Figure 6, top). The overall statistics for all evaluated heats are depicted below for the steel and slag phase.

In particular, the phosphorus content in the steel phase is accurately captured using the described modelling approach. The calculated phosphorus content in the slag also depicts the trend correctly but shows larger deviations than in the steel phase. Considering that the slag phase is less homogeneous, the result is still satisfactory. This result confirms that the selected approach for describing the dephosphorisation reaction in the EAF is suitable and accurate enough to be used for an online monitoring of the phosphorus content evolution.

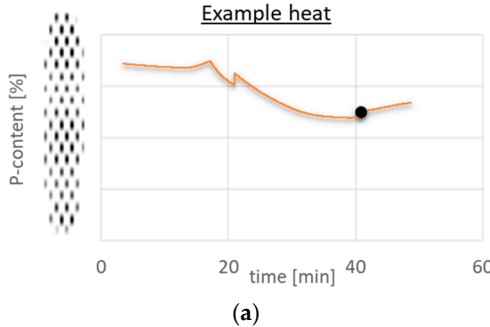

**Figure 6.** *Cont.*

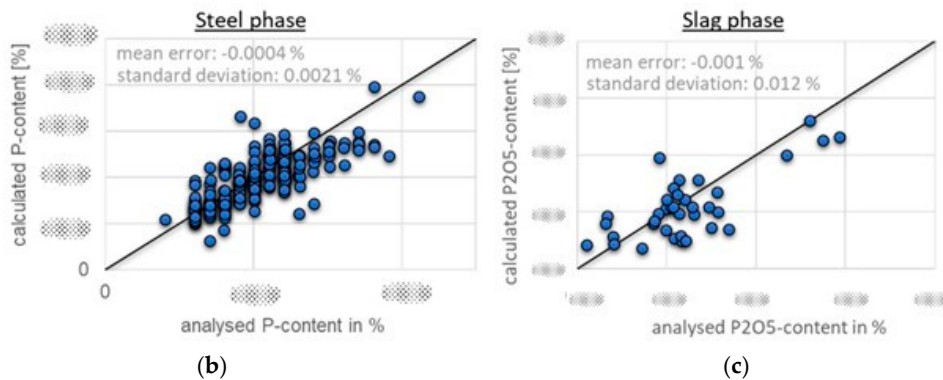

(**b**)　　　　　　　　　　　　　　　　　(**c**)

**Figure 6.** Calculated phosphorus content (orange line) and comparison with measurement (black dot) for one example heat (**a**); model accuracy of phosphorus calculation ((**b**): steel phase; (**c**): slag phase).

### 3.2. Online Implementation for Process Monitoring

The dynamic process model is embedded for the simulation of historical heats within the above-described web-based software tool. For online monitoring, it was implemented within the process control system at Sidenor. A model shell uses the process data stored in a database to initiate the heat state calculations within the model kernel. The calculation results are returned to the model shell and written back into the database in time intervals of one second. To provide the monitoring functionality to the plant operators, a user interface was developed that displays the model input and output data during EAF operation (see Figure 7).

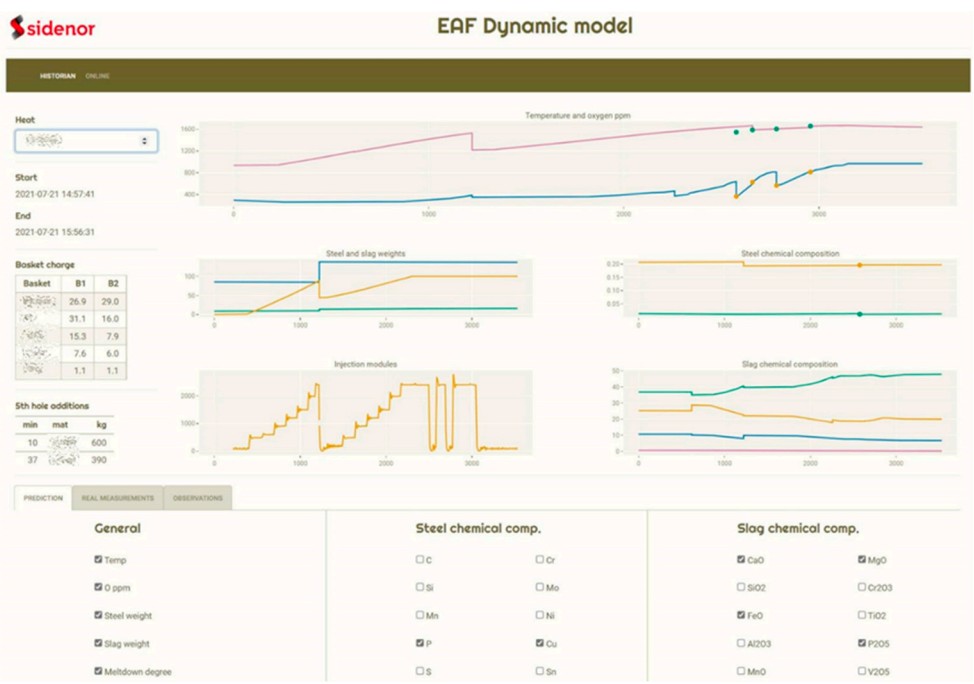

**Figure 7.** User interface for process monitoring with dynamic EAF process model at Sidenor, with curve display of melt temperature and oxygen content (**top**), steel and slag weight (**middle left**), steel composition (**middle right**), oxygen injection rate (**bottom left**) and slag composition (**bottom right**).

Charged materials, current operating conditions and the heat state in terms of melt temperature as well as steel and slag weight and composition are visualized by trend lines. A checkbox menu allows the user to decide which information shall be displayed to improve clearness of the GUI. In addition to the live data, historical heats can also be loaded and visualised in order to analyse the process and model performance from past

EAF operation. Simultaneously, logging files are created on the local machine that are used to evaluate the model accuracy and to optimize the model parameters with the help of offline tools.

The good model accuracy in predicting melt temperature and steel composition was confirmed when evaluating the results achieved with the online implementation of the model. This is exemplarily shown for the melt temperature and the phosphorus content in Figure 8.

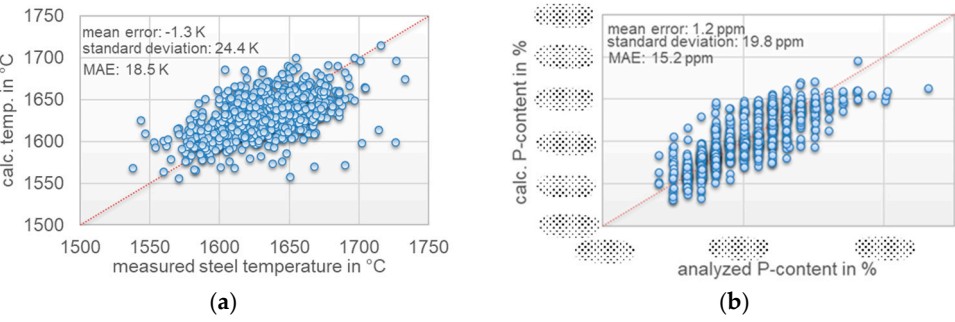

**Figure 8.** Accuracy of model-based online process monitoring regarding melt temperature (**a**) and phosphorus content (**b**).

### 3.3. Model-Based Decision Support

A direct application of the online process model for decision support is for end-point control. The operator can monitor the process behaviour with the evolution of the melt temperature and the most important elements, and on this basis, they can decide when the targets of the process have been achieved and the melt can be tapped from the EAF.

Furthermore, the dynamic process model is also used during the refining phase to predict the further evolution of the melt temperature and important element contents like oxygen and phosphorus, and a model predictive control (MPC) approach proposes optimal control set-points to achieve the desired target values. As a prediction horizon, a time window of 5 min with 10 s time-step intervals has been chosen as a compromise between computation time and sufficient forecasting time for the operator. Regarding the control options, oxygen and carbon injections as well as lime injections were identified as suitable variables. The set-points for these variables are restricted with respect to common operating practices, which yields three possible set-points for the oxygen injection and four for the carbon injection. Possible lime injections are implemented in 100 kg steps up to a maximum value of 500 kg. For the cost function for optimization, the oxygen and phosphorus content of the liquid steel were identified as critical quality indicators. Additionally, the tapping temperature of 1640 °C can be included as a further target value, but it is less penalized than the quality indicators. The changes in control set-points are also penalized to avoid fluctuating control suggestions.

In Figure 9, the result of the online applied MPC algorithm is depicted for an example heat. The upper diagrams show the previously calculated (solid line) evolution of the oxygen content in the refining phase. The red dashed line describes the forecasted evolution based on the currently applied control set-points, whereas the green dashed line is based on optimised control inputs.

It can be seen that the oxygen content stays below the target value when applying the optimized control set-points. The lower part of the figure shows the control set-point values for carbon, oxygen and lime injection. Solid lines indicate the applied set-points, whereas the dashed lines depict the suggestions of the MPC control. It can be clearly seen in the right part of the figure that when applying the optimal set-points, the oxygen content stays safely below the targeted maximum value. This example proves that an MPC approach based on a dynamic EAF process model is suitable for an online control of selected input parameters of the EAF process. The calculation time of the dynamic model is low enough

to allow iterative calculations to be performed to determine the optimal control set-points throughout the refining phase of the EAF process.

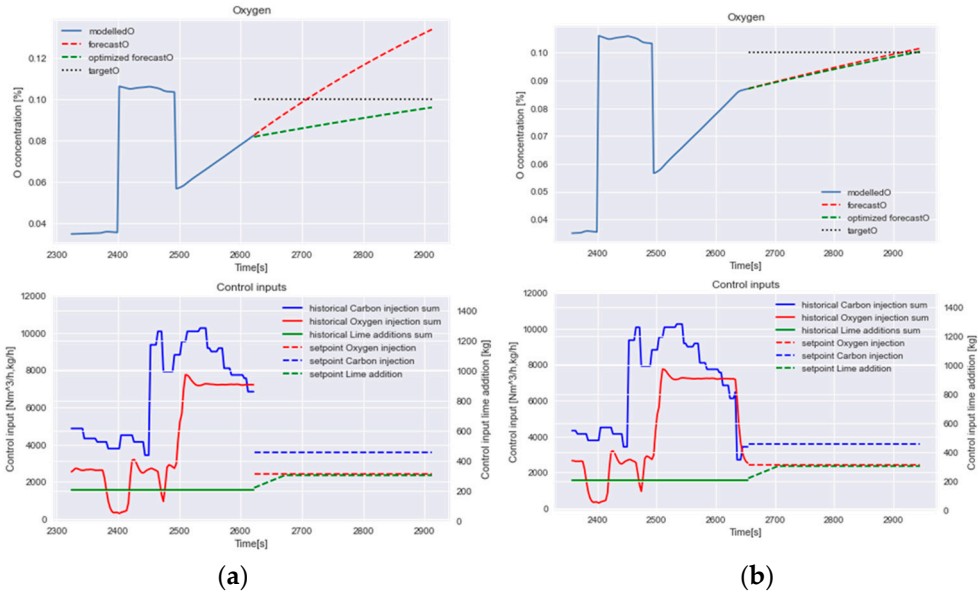

**Figure 9.** Results of MPC control for an example heat, depicted for the oxygen content. Lower carbon and oxygen flow rates are suggested (**a**), which are matched by operator later (**b**).

## 4. Conclusions

A statistical model for scrap characterisation was combined with a scrap mix optimization calculation and a dynamic process model to build a powerful decision support system for EAF process engineers and operators. The online implementation of the model allows the operator to perform accurate monitoring of the process behaviour and can be applied for exact process end-point control regarding melt temperature as well as oxygen, carbon and phosphorus content. Embedded within a model predictive control concept, the model can provide useful advice to the operator to adjust the relevant set-points for energy and resource-efficient process control.

The whole decision support system has been implemented at the EAF plant of Sidenor Aceros Especiales in Spain and is currently being tested in an industrial environment.

The developed decision support system is a relevant contribution towards the digitalisation of the EAF process, covering not only the melting furnace itself but also the scrap yard, with the selection of the optimal scrap mix. So far, mainly classical statistical approaches and analytical process models have been used to set up this system. Currently, several follow-up projects are running whereby methods of artificial intelligence and machine learning will be embedded into the solution [29,30]. This will, on the one hand, further improve the prediction accuracy of the models, but on the other hand, it will also enable the easy transfer of the decision support system to other EAF steel plants.

In the near future, it is expected that the EAF will play an even more important role in steelmaking, due to the demand for decarbonization and circular economy solutions of steelmaking. This underpins the requirement to provide flexible tools for the digitalization of the EAF process.

**Author Contributions:** Conceptualization, W.K. and B.K.; methodology, W.K., B.K. and I.U.I.; software, W.K., D.M.V. and A.A.A.; validation, all authors; writing—original draft preparation, B.K.; project administration, B.K.; funding acquisition, B.K. and I.U.I. All authors have read and agreed to the published version of the manuscript.

**Funding:** This research was funded by European Union's Horizon 2020 research and innovation program within the SPIRE PPP, grant number 869882.

**Data Availability Statement:** Not applicable.

**Conflicts of Interest:** The authors declare no conflict of interest.

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
