# Peer review of "Model-Based Decision Support System for Electric Arc Furnace (EAF) Online Monitoring and Control"

_metals, doi:10.3390/met13081332_

Round 1

Reviewer 1 Report

The article in review is interesting. It is based on the extensive metallurgical experience of the authors, which can be seen in the work. The presented results are based on a "big cloud" of results obtained under industrial conditions. Furthermore, the concept described by the authors was validated. The effects of the work are interesting.

Consider the title of Chapter 4. Discussion and results are unfortunate, as the authors discussed the results by presenting them in earlier chapters.

Here are some detailed notes on the work:

1. Figure 2. Markings a and b should be introduced in the drawings and the description in the drawing should be corrected taking into account the entered symbols a and b.

2. Why are the descriptions given in Figure 4 of the calculated steel temperature (Celsius), measured steel temperature (Celsius), and statistical parameter values using a Kelvin unit?

3. Figure 4 consists of three elements; in order to improve readability, it is proposed to use the notation a, b and c. Furthermore, the caption under Figure 4 should be supplemented with the notation a, b and c.

4. line 189: "An error standard deviation of around 25 K was achieved..." Unit degrees Celsius was used to record the temperature value. Why was it decided to use different units to describe the temperature value and its statistical parameters?

5. Figures 5 and 6 are illegible. Especially the values on the axes. You should also mark figures a, b, and c and correct the description with new symbols.

6. Figure 7 is of poor quality.

7. Figure 8. Please enter the a and b indexes on the drawings. Note as before. Why are the temperature values not visible in the image on the right?

Author Response

Dear Reviewer, 

thank you for the commetns to the paper. Please find below the answers:

Questions 1, 3, 5 and 7: Markings of figure parts have been introduced.

Questions 2 and 4: Temperature differences and modeling errors are normally given in Kelvin, only absolute temperatures are given in °C.

Question 5 and 7: Some figures on the axes were blurred due to confidentiality issues of our industrial partner.

Question 6: Figure has been re-introduced and enlarged, no better quality is available. Please note that also in the screenshot some values are blurred, see above.

Titel of chapter 4 has been changed to "Conclusions" 

Reviewer 2 Report

Comments

It is an interesting applied research and suitable for publication in this journal. Additionaly it is  very important that at this time this usefull tool of support system for EAF is tested in the industrial scale. Before its publication I have some comments below:

-Please give the explanations of abbreviations in the text as EAF, BFI, GUI

- If it is easy please give more references in your text or discuss further the included ones to be more clear your state of the art. Moreover, for the same reason please emphasise more in your text about  the dephosphorisation reaction in order to be strengthened your novelty of your work.

-Do your input data include scrap phase  (mineral) compositions, since they influence the final T of melt?please refer something about this.

Kind regards

Author Response

Dear Reviewer, 

thank you for the commenst to our paper. Please find below the answers.

Abbreviations BFI and GUI were explained, EAF was already explained before.

More references were provided, and the dephosphorisation model has been explained  in more detail.

The model considers also the non-metallic part of the slag as contribution to the slag. This was explained in the text.

Reviewer 3 Report

This is an interesting and well elaborated work. The objectives set by the authors are met and it is possible that the methodology proposed here can be taken to other production plants and consequently lead to improvements in the quality and productivity of the processes in the steel industry.

I consider that some modifications should be made in the wording of the article in order to improve the clarity of the document and to make it suitable for publication. Among them, the following stand out:

- The document should be written in a uniform way and not mix verb tenses. The document should use the passive voice and the active voice of the present and past tenses. No animated actions should be assigned to the paper, figures or tables. Do not say "This work presents...", say "In this work is presented...". Do not say "Figure 1 shows..." say "In Figure 1 is shown...".

- The abbreviations BFI, GUI should be identified.

- Do not present results in the Materials and Methods section. Do not mix information

- It is important that the authors consider the option of presenting a Design of Experiments, in order to give greater reliability to the experimental work.

- Figure 1 should be clarified. What does each type of scrap consist of?

- It would be interesting to accompany the processes studied with the corresponding chemical reactions that occur in the processes studied: i.e. dephosphorization, oxygenation, etc.

- Review line 259. The words should be broken correctly on each line.

- Section 3 should be called Results and Discussion. Not just Results. The discussion should be deeper and more detailed, because in many moments it is only descriptive. I consider that it lacks a little more analysis.

- Section 4 called Discussion and Conclusions is very poor and should be complemented. Therefore, I propose that it should be only Conclusions and space should be given to the recommendations that come out of this study.

- The references presented are very old. There is only one citation to an article from 2020. All other references are more than 10 years old and this should be reviewed. This section should be improved and updated.

Author Response

Dear reviewer, 

thank you very much for the comments to our paper. Please find below the answers.

Phrases are written in a uniform way.

Abbreviations BFI and GUI have been explained

The reason why no DoE was applied was explained.

Figure 1 was commented in more detail.

More details about chemical reactions were given.

Word breaking was corrected.

Heading of chapter 3 was changed to "Results and Discussion", and the analysis was extended.

Conclusions were extended

Reference section was enlarged with much more new references, mainly from 2021.